# The Sex–Gender Effects in the Road to Tailored Botanicals

**DOI:** 10.3390/nu11071637

**Published:** 2019-07-17

**Authors:** Ilaria Campesi, Annalisa Romani, Flavia Franconi

**Affiliations:** 1Dipartimento di Scienze Biomediche, Università degli Studi di Sassari, 07100 Sassari, Italy; 2Laboratory of Sex–gender Medicine, National Institute of Biostructures and Biosystems, 07100 Sassari, Italy; 3Laboratorio PHYTOLAB (Pharmaceutical, Cosmetic, Food supplement Technology and Analysis)-DiSIA Università degli Studi di Firenze, 50019 Firenze, Italy; 4Laboratorio di Qualità delle Merci e Affidabilità di Prodotto Università degli Studi di Firenze, 59100 Firenze, Italy

**Keywords:** sex–gender, phenols, pharmacokinetics, microbiota, pregnancy, lactation, developmental trajectory

## Abstract

Phenols are a wide family of phytochemicals that are characterized by large chemical diversity and are considered to bioactive molecules of foods, beverages, and botanicals. Although they have a multitude of biological actions, their beneficial effects are rarely evidenced in clinical research with high scientific rigor. This may occur due to the presence of numerous confounders, such as the modulation of phenol bioavailability, which can be regulated by microbiota, age, sex-gender. Sex-gender is an important determinant of health and well-being, and has an impact on environmental and occupational risks, access to health care, disease prevalence, and treatment outcomes. In addition, xenobiotic responses may be strongly influenced by sex-gender. This review describes how sex–gender differentially influences the activities of phenols also in some critical periods of women life such as pregnancy and lactation, considering also the sex of fetuses and infants. Thus, sex–gender is a variable that must be carefully considered and should be used to propose directions for future research on the road to tailored medicine and nutrition.

## 1. Introduction

A total of 52 million US citizens use complementary and alternative medicines [1]. Natural compounds are also largely consumed in Asia, Europe, and Latin America (30–40% of the population) [2]. Among them, the use of botanicals continues to increase globally [3]. About 20,000 different botanical formulations are available in the USA market [4] and many of them are widely used in complementary and alternative medicines [5]. The use of botanicals depends on age, education, whether the individual is physically active, and sex [[6] and cited literature].

Rigorous findings regarding the efficacy and safety profile of botanicals are lacking ([6] and cited literature). There are some biases in the research focusing on the health effects of botanicals including phenols, such as their high placebo effect, which often leads to equivocal and non-significant effects ([6] and cited literature). The botanical response is also influenced by individual factors (age, sex-gender, body size and composition, genetics, psychological stressors (loneliness, etc.), social status, instruction, microbiota, lifestyles, etc.). The botanicals activity and the outcomes of preclinical and clinical investigations may be dependent on the above factors.

Here, we focus on phenols, secondary metabolites of plants, and sex and gender in consideration of the fact that women are the biggest users of botanicals ([6] and cited literature).

There are different definitions of sex and gender [7,8,9,10]. According to the World Health Organization, the terms “gender” or “sex” refer to two different concepts. Sex “refers to the set of biological characteristics that define humans as female or male”; whereas gender “refers to the socially constructed roles, behaviors, activities, and attributes that a given society considers appropriate for men and women”. Evidently, it is easier to study sexual dimorphism, because sex is considered as a dichotomous variable; while gender is a much more complex because it implicates socio-cultural and political constructions. Sex and gender influence each other through complex and continue interactions; thus, sometimes it is impossible to distinguish between the two concepts and, therefore, we, as others have [11,12], prefer to use the mixed term, sex–gender.

Recently, the biological activities of phenols are reviewed [13,14,15,16,17,18], but often, the sex–gender aspect is not appropriately studied, with some exceptions [6,19,20]. Indeed, despite a large amount of information regarding the biological activity of phenols, clear evidence regarding their effect on human health is lacking [13,14,15,16]. Beyond individual factors, their response is largely dependent on their chemical structure, pharmacokinetic properties of the single phenol. Here, we focus on critical periods of a woman’s life, namely gestation and lactation, considering the influences of sex of the fetus. In addition, starting from “Thrifty Phenotype Hypothesis” [21], the early and tardive effects of sex–gender of the fetus on phenol exposure in pregnancy and lactation are discussed.

## 2. Plant Materials

Botanicals are often complex mixtures and can present a great variability [22] depending on cultivar, the country of cultivation or spontaneous growth (altitudes, climates), the season of harvest [23,24], and practices of manufacturers [25]. Thus, it is not surprising that Simpraga et al. [26] defined the plants as “chemical factories”, because they generate numerous compounds. Primary metabolites play a role in basal functions, whereas secondary ones are involved in the defense against numerous stressors [22,27].

Plants produce a large number of phenols (>8000), but edible plants produce only several hundred of phenols [28]. Chemically, phenols are classified and categorized as flavonoids (isoflavones, flavones, flavanols (catechins and proanthocyanidins), flavanones, anthocyanidins and flavonols, and non-flavonoid compounds (phenolic acids, stilbenes, lignans). Flavonoids have two aromatic rings linked through a three-carbon chain (Figure 1). Tannins are oligomers and polymers of flavonoids and are classified into two groups, condensed tannins and hydrolyzable tannins.

High amounts of coumestans and stilbenes are measured in clover and in cocoa- and grape-containing products, respectively [29]. High amounts of isoflavones (genistin, daidzin, and glycitin) and lignans are present in soybean and in flax seed, respectively [29]. Formononetin and biochanin A are present in the red clover and are precursors of genistein and daidzein, respectively [6]. High amounts of isoflavones and lignans are present in soy bean and in flax seed, respectively [29]. In olive fruit and in virgin olive oil are present numerous phenols among them 5-hydroxytyrosol (HT) and oleuropein derivatives are noteworthy [30]. Notably, humans may produce tyrosol (Tyr) and HT as minor metabolites in dopamine and tyramine metabolism [31]. In plants, phenols are mainly presented as glycosides ([6] and cited literature).

## 3. Sex Differences in Plants

Plants are complex entities that, in part, depend on natural selection and evolution [22]. While the stress responses of plants have been widely investigated [32], studies on the influences of sex on stress response are scarcer. Recently, it has been shown that the sex of a plant may influence phenol production. Leaves collected from *Pistacia atlantica* vary with regard to their contents of total phenols, flavonoids, and condensed tannins depending on the month they were harvested, their growing region, and sex, whereas their antioxidant properties are not influenced by the sex of plant [33]. In *Salix myrsinifolia*, the amount of phenols is enhanced by prolonged exhibition to UV and this enhancement depends on the sex of the plant. In particular, UV radiation reduces the activities of phenol oxidase, especially in male plants, whereas guaiacol peroxidase activity is not affected by UV radiation and sex [34]. Male *Populus cathayana* Rehd are more protected from UV than females. Accordingly, males possess a greater antioxidant capacity and higher anthocyanin levels than females [35]. Whereas in *Ginkgo biloba* and *Rhodiola rosea* L., the phenolic amount and the defense versus oxidant do not diverge in male and female [36,37].

This point requires further investigation because it could be a criterion in choosing the sex of the plant in order to use the right sex to obtain more phenols.

## 4. Pharmacokinetics of Phenols: Influence of Sex–Gender

Bioavailability is an essential aspect when investigating the phenols’ effects in vivo. Only 1–10% of the phenol assumption can be detected in plasma and urine, where phenol contents range from nM to low µM [16]. Anthocyanidins and flavones have a low bioavailability, followed by flavanols, flavanones, and soy isoflavones. Glycosides and aglycones have different levels of bioavailability. However, glycosides may be metabolized in the gut, contributing to the high variability in phenol bioavailability because the combined activity of different enzymes of the host and of microbiota and the different transporters [38].

There are data supporting the importance of ethnicity in the bioavailability of phenols. Circulating daidzein, genistein, and equol are higher in Japanese men and women (>40 years old) versus British men and women. In addition, more Japanese men and women are able to produce equol versus English men and women. Finally, enterolactone has the same serum concentration in Japanese and English cohorts [39].

Importantly, the sexual dimorphisms that can affect the phenol kinetics are reported in Table 1.

### 4.1. Absorption

The oral bioavailability of exogenous compounds depends on the characteristics of individuals (stomach pH, gastrointestinal mobility, enzymatic degradations, mucus secretion, transporters, microbiota, age, etc.) and these often present a marked sexual dimorphism (Table 1 and Table 2) and by the chemical structure of the molecule, plant matrix, and the formulations. 

Generally, the human stomach is of low importance for absorption. Indeed, black raspberry anthocyanins are one of the few compounds that can be absorbed in the stomach [43].

The most important location for absorption is the intestine. The absorption mainly occurs by passive diffusion, although some phenols may be transported by the sodium-glucose co-transporter 1, which at renal level diverges between male and female rats ([19] and cited literature). The absorption also depends on sugar, because it modulates the hydrolysis of glycosides by lactase phlorizin hydrolase and by gut micro-organisms, which produce by β-glucosidases ([19,44] and cited literature). 

Phenol absorption may also be regulated by several transmembrane proteins, such as P-glycoprotein (P-gp), breast cancer resistant protein (BCRP), peptide transporter (PEPT), and organic anion transporting polypeptide (Oatp) [45]. BCRP excretes phenol metabolites into the gut, thus decreasing the bioavailability of phenols participating in their recycling [46]. This is of special value for glucuronide and sulfate conjugates, which are more hydrophilic than their precursor and, therefore, the efflux transporter is vital to facilitate their exit from the cells [46]. Notably, BCRP is more expressed in the liver of male rats and men versus the liver of female rats and women [47]. Genistein, daidzein, and many of their sulfated conjugates are transported by BCRP; with genistein being more absorbed in female rats versus male rats [48]. Finally, others show that the bioavailability of genistein and of its aglycone is much higher in female compared to male rats and the authors suggest that it depends on the recycling process [49]. Recently, it has been shown that isoflavones are inhibitors of these transporters [48].

Considering the sex differences present at gastrointestinal level (Table 1), it is plausible that sex differences are present in phenols gastrointestinal absorption. 

### 4.2. Metabolism

The phenol biotransformation begins in the mouth (Figure 2), where the phenols interact with saliva salivary glands secretions, epithelial cells, micro-organisms, and both cell types produce β-glycosidases ([19] and cited literature). However, the real value of the oral transformations is unclear due to the short time that most foods and/or supplements remain in the mouth. However, aglycones of black raspberry anthocyanins could be detected in the mouth, and the saliva contains glucuronidated anthocyanin conjugates [43]. Importantly, oral phenol metabolism may be influenced by female sexual hormones, suggesting that mouth microbiota in women may vary with fertile status [50]. In addition, hormone replacement therapy may change oral microbiota but these are not univocal data ([50] and cited literature). Importantly, oral microbiota seems to change during pregnancy [51] and this could have some consequences on phenol metabolism. 

The bulk of phenols are metabolized in the intestine where they are hydrolyzed by lactase phlorizin hydrolase and by β-glucosidases of gut microbiota [52]. Unlike humans, rats and mice of both sexes produce equol similarly [53], indicating that phenol metabolism is species-specific. Additionally, the cytochrome P450 (CYP) enzymes, uridine diphospho-glucuronosyltransferase (UGT), sulfotransferase (SULT), N-acetyltransferase (NAT) and glutathione S-transferase (GST) and catechol-*O*-methyl transferase (COMT) are involved in phenols metabolism (Figure 2) [54].

Relevantly, CYP present numerous sexual dimorphisms (Table 1). CYP activities is under the control of constitutive androstane receptor (CAR), pregnane X receptor (PXR), peroxisome proliferator-activated receptor α (PPARα), and aryl hydrocarbon receptor (AhR) [19]. CAR regulates genes of CYP2B6, CYP2C8/9, and CYP3A4, UGT, and some carriers such as multidrug resistance-associated proteins 2 and 3 ([55] and cited literature). In women, CYP2B6 activity is higher compared to men, although there are no univocal data [56,57]. CYP2B6 is more elevated in Hispanic women versus white or African–American women [19]. In addition, CYP2B6 shows a sex difference in its genetic polymorphisms, with 1459C > T SNP and intron-3 15582C > T SNP associated with a very low level of CYP2B6 in females [41]. CAR regulates the dimorphic induction of the CYP2B1 gene (which is lower in female than in male rat livers). Interestingly, sexual male hormones appear to inhibit CAR [19], whereas estrogens activate it [58]. Furthermore, the activity of CAR is influenced by pregnancy [59].

PXR regulates numerous CYP (CYP3A, CYP2B, CYP2C, CYP2A6), and several other enzymes (UGT1A1, UGT1A9, UGT1A3, UGT1A4, UGT1A6, GST, SULT) and some organic anion transporters [60]. A rich-fat diet reduces protein expression of estrogen receptor *α* (ER*α*) in hepatic and white adipose tissue in transgenic female mice expressing the human PXR gene [61]. Phenols may regulate the activity of PXR ([19] and cited literature). Importantly, PXR plays a role in the variation of CYP enzymes observed in pregnancy [62]. 

PPARα activates both numerous CYP such as CYP2C8, CYP3A4, CYP7A1, CYP8B1, and other enzymes such as GST and UGT1A9 [63]. Importantly, CYP3A4 is also induced via PXR and CAR [64]. The expression of PPARα is higher in the male liver and in the activated male T cells compared to in female ones [65,66]. After fasting, PPARα protein and mRNA are more expressed in the hepatic tissue of male rats versus female rats. In detail, PPARα mRNA increases by 72% and 52% in males and in females, respectively [66]. This sexual dimorphism is not evident after hypophysectomy. Furthermore, in lymphocytes, PPARα mRNA and protein are more marked in male mice versus female ones [67]; however, T cells function and immune virulence are altered by PPARα only in males [67].

AhR activity depends on species, strain, sex, age, tissue, and cell-specific responses. It is activated by quercetin and resveratrol and its derivates in primary human hepatocytes [68,69]. Biochanin A and formononetin seem to be AhR agonists and may induce CYP1B1 and CYP1A1 [6] Elevation of placental CYP1A1 activity, which is mediated by AhR, is linked with premature birth and other fetal alterations in smoking women [70].

Some flavones (chrysin, baicalein, and galangin), naringenin and isoflavones (genistein, biochanin A) are able to inhibit the activity of aromatase CYP19, which has a pivotal role in the production of estrogens, reducing their synthesis [71]. Rutin, myricetin, isorhamnetin, p-coumaric acid, gallic acid, and caffeic acid exert a sex-specific inhibition on CYP1A, CYP2A, CYP2E1 (Table 2), and CYP3A being higher in female pigs than in male ones ([19] and cited literature), whereas quercetin-induced inhibition of CYP2E1 and myricetin-induced inhibition of CYP3A are present only in male pigs ([19] and cited literature) (Table 2). The activity of CYP2D2, which is higher in male rats versus females, and that of CYP3A is inhibited by resveratrol) and isorhamnetin in both sexes ([19,72] and cited literature) (Table 2). In men and women, the sexual dimorphism is not as evident ([19] and cited literature).

Both in humans and in non-human animals, some foods and beverages, such as *Brassicaceae* vegetables, soya derivate products, citrus fruits, etc., induce UGT ([19] and cited literature). The activities of phase II enzyme may be influenced by sex–gender, with disulphates, followed by 7-sulpho-4′-glucuronides derivates being the main metabolites in male rats, whereas female rats produce more 7-glucuronides ([19] and cited literature). After ingestion of olive leaf extract or olive oil, the plasma concentration of HT glucuronide, HT sulfate, and oleuropein aglycone glucuronide are higher in postmenopausal women versus premenopausal women [73,74]. Plasma genistein glucuronide levels are higher (>2 fold) in female than in male Sprague Dawley rats, and are also higher versus genistein aglycone in females but not in males, although these data are not univocal [48]. Notably, some glucuronides of some phenolics (e.g., daidzein and genistein) retain certain biological activities. In human hepatic microsomes, the conjugation through UGT1A1 and UGT1A3 of resveratrol and pterostilbene is more efficient in females than in male microsomes [75]. Glucuronides are more hydrophilic than their precursors, thus the passage from intracellular to extracellular compartments needs transporters, building an interplay between UGT and efflux transporters. The COMT, which is more expressed in male individuals, [19] is inhibited, in vitro, by epigallocatechin-3-gallate (EGCG) [76]. However, a single dose of EGCG does not impair the COMT of red blood cells in healthy individuals [77]. A synthetic inhibitor of COMT (tolcapone) reduces the seeking or consumption of ethanol, specifically in male high drinkers [78]. Importantly, phenols can regulate the ratio between CYP enzyme/phase II enzymes, which can be a pivotal phenomenon for the life and death of cells after carcinogens [16]. Among CYP, UGT, and phenols, there is a bidirectional interplay because phenols may regulate the activity of CYP and UGT as they are also metabolized by them.

Some variability in phenol metabolism could depend on sexual hormones [48], and by microbiota (see below). 

### 4.3. Distribution

After absorption, a phenol is distributed into interstitial and intracellular fluids and tissues. The distribution depends on molecules (lipid solubility, dimension, etc.) and individual characteristics (vascular permeability, regional blood flow, and perfusion plasma proteins bindings, etc.). Phenols binding to albumin or to other plasma proteins is an important determinant in phenols availability and distribution. Information about sex–gender differences in human serum albumin levels is scarce, although circulating levels of albumin are influenced by age, sex, and pregnancy [79]. Phenols may bind serum proteins, and this may modify phenol distributions and induce phenol–drug interactions. For example, casticin and resveratrol slightly displace naproxen from human albumin [80]. In vitro, acacetin, apigenin, chrysin, luteolin, galangin, quercetin hesperetin, and naringenin aglycones may inhibit the binding of warfarin [81]. Unfortunately, it is not known if the binding of phenols with protein presents sexual dimorphisms or if it changes with pregnancy. In view of human sexual dimorphism presented in Table 1, it is plausible that sex–gender differences in phenol distribution exist. For example, the administration of grape seed phenols presents a different distribution in plasma, brain, and liver of female and male rats [82] and that may affect the activities of flavanols in males and in females.

### 4.4. Elimination

The conjugated forms are excreted in the urine [83]. After absorption of olive oil, kidney elimination of HT and that of its major microbial metabolites homovanillyl alcohol are higher in humans than in rats [84], indicating that the kidney elimination is species-dependent. The ingestion of HT derivatives modifies the kidney elimination of dopamine, noradrenaline, normetanephrine, and 3-methoxytyramine, with the effect being larger in males (12-fold) than in females (1.5-fold), indicating that interaction among phenols and catecholamines occur [85]. Human HT excretion is also influenced by formulation [84]. Finally, urinary excretion of phenols depends on sex–gender and on age. In an elderly Italian population (more than 56% have impaired renal function), the total urinary excretion of phenols decreases with age and is greater in men than in women [86]. Upon the commencement of daily consumption of soy milk, urinary elimination of isoflavone conjugates is larger in women than in men; however, only in women there is a progressive reduction in excretion of genistein and daidzein [87]. In fact, it is not known if urinary microbiota influences the phenol elimination in a sex–gender-specific way. This could be plausible because adult men and women have different urinary microbiota [88]. The female microbiota depends on sex hormones. A large abundance of *Clostridia* and *Ruminoccocaceae* genera is linked with urinary estrogens [89]. Intriguingly, the bladder microbiota of pregnant women resembles those of perimenopausal women [88]. It varies in overactive bladders with and without urgency urinary incontinence [88], and in major depression [90], a mood disturb that mainly affects women versus men [91].

Phenols can also be eliminated in the feces or deconjugated by gut microbiota (see below and Figure 2), which results in their reabsorption and enterohepatic recirculation [83]. Finally, they can be eliminated as CO_2_ by the lungs [83].

## 5. Microbiota

The intestinal microbiota is a complex and dynamic ecosystem that strictly cooperates with the host to maintain host homeostasis and it depends on numerous environmental (diet, drugs, etc.) and individual factors, including sex–gender; thus, it is highly personalized by life experiences ([19,92] and cited literature).

Healthy women have higher intestinal microbial diversity and a relative abundance of specific taxa versus healthy men [93,94]. Interestingly, age has different effects in males versus females, with the age effect on *Streptococcus salivarius* being female-specific [95]). The sex–gender diversity of microbiota also depends on body dimensions. The abundance of the *Bacteroides* is lower in obese men than in obese women, whereas *B. plebeius* is higher only in obese men. Possessing a body mass index (BMI) of over 33 changes the *Firmicutes/Bacteriodetes* ratio [94]. Women and men with a BMI < 33 do not present significant differences. The gut microbiota diverges in premenopausal and postmenopausal women, the latter have a gut microbiota that is very similar to that of men [96]. In postmenopausal women, genistein and glycitin modify the intestinal microbiota by elevating and lowering the concentration of the *Bifidobacterium* and *Clostridiaceae*, respectively [[19] and cited literature]. Bilateral ovariectomy causes dysbiosis, increasing the abundance of *Clostridium bolteae* [95]. Gut microbiota composition in women is also influenced by oral contraceptives, which alter both microbial species abundance and functional pathways [95]. Not only do estrogens impact on microbiota but microbiota itself may regulate circulating estrogens [97]. The bacterial β-glucuronidases deconjugate a variety of endogenous molecules and numerous xenobiotics, including sexual hormones and phenols [98]. Starting from flavonoids, humans produce equol, enterolactone, and enterodiol ([52] and cited literature). S-equol is produced only by some subjects ([52] and cited literature), whereas other individuals, who consume soy, may produce O-desmethylangolensin via 2′-dehydro-*O*-demethylangolensin. Moreover, *Clostridium* and *E. ramulus* species induce formation of flavanones aglycones and further degradation in the colon ([52] and cited literature). The gut microbiota, through *Bacteroides* and *Clostridium* species, transforms lignans into estrogenic enterolactone and enterodiol ([52] and cited literature). *Clostridium scindens* convert glucorticoids to androgens [99]. Blueberry supplementation (rich in anthocyanins and phenolic acids) alters gut microbiota, increasing the *Bacteroidetes* and decreasing the *Firmicutes*, with the effect being greater in females than in males. Beside the five genera unique to males, *Clostridium, Corynebacterium*, and *Facklamia* are elevated, while *Ruminococcus* and *RF39* are lowered. In female mice, *Adlercreutzia*, *Coprococcus*, *Mogibacteriaceae*, *Turicibacter*, and *S24-7* are elevated, while *Anaerotruncus*, *Christensenellaceae*, *Ruminococcus*, *Mucispirillum*, and *Staphylococcus* are lowered in the blueberry-fed groups [100].

A marker of gut dysbiosis, trimethylamine N-oxide (TMAO) is negatively related with a Mediterranean Diet in a way characterized by an evident sexual dimorphism [101]. Dietary components present in *Cruciferae* inhibit the synthesis of TMAO through inhibition of the flavin-containing monooxygenase form 3 [102], whose expression depends on sexual hormones [102,103].

The above results indicate that a bidirectional interaction co-exists among phenols and the intestinal micro-organisms. 

Gut, vaginal, and oral microbiota changes during pregnancy [104], but it is not known if they change the pharmacokinetics of phenols.

Globally, the microbiota is important and relevant in the biotransformation of phenols and their metabolic products, in turn, can favor the growth of beneficial gut inhabitants, able also to reduces the growth of microbiological agents that can produce dysbiosis ([19] and cited literature). In this context, we recall that numerous phenols have antimicrobial activity, and thus, this may modify their own bacterial metabolism. Therefore, phenols and their metabolites could positively modulate intestinal bacterial populations ([52] and cited literature]). Less it is known about the importance of urinary microbiota in phenols’ pharmacokinetics.

In conclusion, the biodiversity and richness of, at least, intestinal microbiota, are crucial for human health in consideration of the possible beneficial effects of adsorbed vegetable foods and beverages as well as of botanicals including phenols. 

## 6. Interactions

It is hard to make a diagnosis of a supplement–drug interaction because most patients under-report the consumption of nutraceuticals to their physicians [105]. This is especially of interest for women, because they use more supplements and drugs than men [106,107]. However, these are not univocal data because in inpatients, potential interactions are linked with the male sex [108], and are also linked with older age and elevated consumption of supplements and/or drugs [108].

Supplement–drug interactions are mainly pharmacokinetic and most of them are linked with CYP and UGT because they metabolize more than 90% of prescribed drugs [109]; however, drug uptake/efflux transporters, and albumin binding are also involved [45]. Relevantly, 70% of drugs are metabolized through CYP3A4, that it is higher expressed in females; thus, the induction or inhibition of CYP3A4 can occur to lower efficacy or to toxic effects of numerous pharmacological medications, including oral contraceptives [110]. St John’s wort was banned by the French Ministry of Health [111] because it may interact with antidepressants, potentially causing fatal serotonin syndrome [112]. Considering that depression is more common in women [91], this could aggravate the risk in women who take St. John’s wort. Indeed, there are also pharmacodynamic interactions resulting either in a decrease or increase in response. The most studied pharmacodynamic botanical–drug interactions regards antithrombotic and anticoagulant drugs [113]. Indeed, in healthy individuals, flavan-3-ol-enriched dark chocolate reduces postprandial platelet function versus standard dark chocolate, in a sex–gender-dependent way; having a major effect in men than in women [114].

Finally, after indomethacin, intestinal permeability increases in both sexes, but alteration in fecal bacterial diversity is present only in women. After discontinuation of indomethacin, these changes are reversible [93]. At the moment, there is lower awareness of the importance of microbiota as a site of phenol–drug interactions. However, these interactions could be very frequent and may occur in a sex-specific way, thus the new term: “pharmacomicrobiomics” [92] appears to be highly appropriate. 

## 7. Development: Pregnancy, Lactation, and Developmental Trajectory

It is recognized that early life (prenatal and neonatal) events have a major role in shaping disease risk (obesity, diabetes, and cardiovascular disease, etc.) in adulthood [21] and this seems to occur, at least in some cases, in sex-dimorphic ways ([115] and cited literature). In particular, the expression levels of the DNA methyltransferases (DNMT) are sexually divergent from blastocysts to adult life. DNMT 1, 3a, and 3b are down-regulated by progesterone ([115] and cited literature), whereas intra-hippocampal infusion of estradiol elevates DNMT3a and 3b expression and also alters histone deacetylase ([115] and cited literature).

### 7.1. Pregnancy

The use of prescription drugs in pregnancy generated two great tragedies (thalidomide and diethylstilbestrol) [116], and this creates a great fear regarding prescription drugs that leads women to use “natural and safe” alternatives, as testified by the large number of pregnant women that use botanicals. In Australia, the prevalence ranges from 14% to 57% [117], whereas it is around 57.8% in the UK [118]. In Italy, the following botanicals chamomile, licorice, fennel, aloe, valerian, echinacea, almond oil, propolis, and cranberry are the most widely used [119], whereas in the USA, the most used herbs are ginger, cranberry, valerian, raspberry, and chamomile [120]. Licorice supplements are also extensively used in Denmark [121]. However, studies on natural alternatives in pregnancy and lactation are few [122]. Thus, much of the information has very little scientific support. Consequentially, the efficacy and safety profiles of phenols are alarming [123]. In this context, one should know if the individual phenol crosses the placenta (genetically it has the same sex as the fetus), the rate and amount of phenols that reach the fetus, the duration and timing of exposure, and the distribution characteristics in different fetal tissues. In addition, herbal products are often co-administered with prescription drugs, raising concerns about possible interactions [124].

Pharmacological response in pregnancy may vary in consideration of the numerous physiological changes induced by pregnancy. Briefly, it is observed a reduction of blood albumins, and an increase in total body water and plasma volume ([125] and cited literature), affecting the distribution. A reduction in gastrointestinal motility is also observed, as is an increase in gastric pH (impacting absorption). In addition, the glomerular filtration rate is increased (impacting renal elimination). Changes in the activity of drug metabolizing enzymes are also described and these changes together with placental enzymes influence the metabolism ([125] and cited literature).

Numerous phenols cross the placenta [19,126,127], and some data suggest that the fetuses tend to retain them longer than the mothers do [126]. Importantly, soy intake elevates daidzein and genistein in amniotic fluid in a dose-dependent manner. This is especially true in the amniotic-fluid-containing female fetuses [[128] and cited literature]. This suggests a possible differential metabolic handling of phenols in male and female fetuses [128]. 

Analysis of 7928 boys, born from mothers that consume a vegetarian diet rich in phytoestrogen, indicates that the vegetarian diet is associated with an increased risk of hypospadias [129]. In women, a high dietary isoflavone intake may reduce fertility [130]. Paradoxically, the use of dietary phytoestrogens is positively related to live-birth rates in assisted reproductive technology [131]. An association between high serum isoflavones levels and the risk of central precocious puberty has been observed in Korean girls [132].

The consumption of fenugreek (*Trigonella foenum graecum*) in pregnancy may induce congenital malformations and death [133,134]. In male rats, mice, and rabbits, fenugreek may induce testicular toxicity and anti-fertility effects, while it alters fertility, implantation, and abortions in females [134]. Prenatal exposure to St. John’s wort increases fetal malformations [135]. Licorice may induce the so-called mineralocorticoid-excess syndrome, inhibiting 11-hydroxysteroid dehydrogenase type 2 [121]. Licorice use may contribute to an increased risk of preeclampsia [136], miscarriage [119], and preterm birth [137]. It is relevant to recall that the side effects of licorice are more frequent in non-pregnant women than in men [138].

Ginger (*Zingiber officinale*) is used to treat nausea and also has prokinetic effects. It contains gingerols and shogaols, which exert anticholinergic activity (muscarinic receptors M3) and antiserotonergic receptor versus 5-hydroxytryptamine receptor 3 [6]. The metabolite, 6-gingerol, may interact with CYP3A4 and CYP2C9 [139], suggesting potential drug–herb interactions. Ginger seems to enhance testosterone production [140] and may alter fetal testosterone metabolism [141]. In addition, ginger seems to decrease platelet aggregation; however, these are not univocal data [142]. If so, inhibition of platelet aggregation may elevate the probability of bleeding in the postpartum.

In line with previous observations, low maternal adherence to a Mediterranean diet during pregnancy is associated with DNA hypomethylation at birth in female babies [143]. Interestingly, some phenols such as EGCG may modify the ER activity modifying DNA hypomethylation and histone deacetylation [144], whereas curcumin inhibits histone deacetylation. Flavonoids, in particular isoflavones, flavonols, and catechins, are regulators of the epigenome through DNA methylation, histone acetylation, and chromatin alterations [144]. Globally, the above data suggest that maternal exposure during pregnancy may affect the epigenome.

The side effects of botanicals are also present after topical administration. In particular, topical administration of aloe or almond oil in the prevention or reduction of striae gravidarum may produce a rash and itching [119]. In almond oil users, miscarriages, preterm labors, and pre-term birth are more frequent than in non-users [119,145]. Finally, microbiota changes throughout pregnancy show that the bacterial diversity is more pronounced during the third trimester ([19] and cited literature), indicating the need to consider it during studies.

In view of the impressive number of plants used during pregnancy and the lack of knowledge about this issue, there is an urgent need to rationalize the use of botanicals in pregnant women considering the sex of the fetus, pharmacokinetics changes, and their potential clinical impact. Immediate attention of regulatory authorities is required because the use of botanicals during pregnancy continues to be a great concern for both women and babies’ health.

### 7.2. Lactation

Breastfeeding women might use prescription drugs or botanical products for postpartum conditions and to induce milk production [146]. In many countries, the use of herbal galactagogues is passed down from generation to generation [146]. Among lactating women, dietary/botanical supplements have not been well-studied. Therefore, many more clinical trials are essential to expand our current knowledge of these products and to develop recommendations [146], for the safety of infants (in consideration of their sex), regarding their potential effects on quantity and quality of milk [147].

### 7.3. Galactagogues Based on Herbs

There are numerous “natural” Galactagogues based on herbs [148] and they are perceived as safe ([6] and cited literature). Despite the use of herbal galactagogues (*Trigonella foenum*, *Foeniculum vulgare*, *Anethum graveolens*, *Pimpinella anisum*, *Nigella sativa*, *Vitex agnus-castus*, *Cnicus benedictus*, *Silybum marianum*, (*Galega officinalis*), *Coleus amboinicus Lour*, *Althaea officinalis*, *Urtica dioica*), scant clinical evidence exists regarding their effectiveness and safety profile [149]. As much as 45% of nursing mothers who used fenugreek reported adverse reactions [150], with diarrhea and hepatomegaly being the most frequent [151]. The use of *Galega officinalis* is not validated by any well-conducted trials; however, its use might cause hypoglycemia [152].

### 7.4. Breast Milk

Human milk is characterized by the presence of a number of factors that are biologically active such as microbes and antimicrobial molecules, functional fatty acids, hormones, oligosaccharides, and stem cells etc. Its composition changes among mothers and during lactation and is also influenced by maternal factors (for instance, diet, stress, BMI) and infant factors, such as sex, suggesting that there are sex-specific requirements during lactation for optimal development ([153] and cited literature). Interestingly, opposite-sex twins are smaller than same-sex twins when breastfed, as maternal milk cannot simultaneously be tailored for both sexes ([153] and cited literature). Moreover, mothers of male babies produce a more caloric milk (25%) than mothers of female infants [154]. This could be attributed to greater growth rates of male versus to female babies. However, in Northern Kenya, economically self-sufficient mothers breastfeed sons more frequently and produce breast milk with a higher fat concentration for sons than for daughters. Vice versa, poor mothers breastfeed daughters more frequently than sons and produce a milk with a more elevated fat amount for daughters compared to sons [155]. These results stress the relevance and importance of social factors on milk production and composition.

Botanical compounds must reach human milk to affect babies. Some phenols such as quercetin are found in the milk [156], whereas *s*oy isoflavones are not effectively transferred in the human milk ([157] and cited literature). These findings demonstrate the importance of chemical structure to be transported into the milk. BCRP plays a relevant role in milk composition participating in its bioavailability and actively affecting the secretion of endogenous molecules and xenobiotics [158]. As already mentioned, some phenols (extracts of soybean, *Gymnema sylvestre*, black cohosh, passion flower, and rutin) strongly inhibit BCRP, while chlorella, milk thistle, and Siberian ginseng extracts are weak inhibitors [159]. Coumestrol is the most potent inhibitor among the tested isoflavonoids [159]. Consequentially, phenols may greatly elevate the availability of BCRP substrates. Finally, the composition of milk microbiota is sex-specific [160], but these are not univocal data [161]; the presence of numerous confounders such as the newborn feeding pattern, domestic water, use of antibiotics in pregnancy and etc. can all affect the composition of milk microbiota (Figure 3) [162]. Interestingly, human milk microbiota may function as a metabolic adaptor. For example, it may compensate for the human milk protein decline, increasing the abundance of bacteria that have genetic software to synthesize amino acids [162].

In conclusion, the maternal milk is the best choice for the feeding of infants as its composition fits with the specific needs of each particular infant (gestational age, time of the day, geographical location, environment, etc.) in a sex-specific manner. This ability indicates that maternal milk is a personalized food source [163].

### 7.5. Soy-Milk Formula

Development is one of the most critical periods for endocrine-disrupting exposure. Thus, certain concerns about the safety profile of soy-formula are not surprising. Soy-milk formula should be an alternative to bovine-milk-formula for infants, which have milk allergy or lactose intolerance. However, the consumption of soy-formula in the USA is estimated to be about 25% of the formula market [157]. The American Academy of Pediatrics, the European Society for Pediatric Gastroenterology Hepatology, and the Nutrition Committee on Nutrition recommend the consume of soy-formula only in true milk allergies or lactose intolerance [157]. The US National Toxicology Programme’s Istance on soy-milk-formula [164] sustains that there is a ‘minimal concern for adverse developmental effects’. However, it is important to recall that, in humans, the knowledge of the tardive effects of soy-milk is limited and of poor quality [157]. Soy-formula intake is linked with altered age at menarche [165,166]. However, data on menarche age are not univocal ([157] and cited literature). A recent clinical retrospective cohort trial found that young women fed with infant soy-formula report longer menstrual bleeding and more menstrual discomfort than those who were fed with a non-soy based formula ([157] and cited literature). Soy-formula intake seems to be a risk factor for uterine fibroids [167] and endometriosis ([157,168] and cited literature). A recent trial reports estrogenized vaginal epithelium in females fed with soy-formula in their first months of life ([157] and cited literature), but other studies do not find a link between soy-infant-formula and developmental reproductive parameters ([157] and cited literature). Notably, a study on mice reports that the neonatal administration of genistein at 18 months increases the incidence of uterine adenocarcinoma, suggesting that genistein exposure during critical periods of differentiation may be cancerogenic [169]. The dose of isoflavones ingested by neonates fed with soy-formula is very high, reaching plasma levels even higher than those reported for Japanese men fed with a soy-based diet [170,171] and exceed 13,000–22,000 times their own endogenous estrogen ([157] and cited literature), altering reproductive hormones [172]. Soy-formula fed infants have higher urine concentrations of isoflavones than cow-milk-formula-fed infants [173]. In contrast, infants fed with cow’s–milk-formula or human breast milk have low plasma isoflavone levels ([157] and cited literature).

The soy-fed babies’ milk microbiota diverges in comparison with human milk and cow-formula-fed babies because they have less *Bifidobacteria* [162]. In fact, at 3–4 months of age, Caucasian male and female infants born from asthmatic women have fewer gut *Lactobacilli* and *Bacteroidaceae* [174]. Formula-fed babies harbor bacteria capable of making a very different set of amino acids that are scarce in formula [162]. Breastfeeding formula ingredients and maternal gestational weight gain have specific effects on gut microbiota [162]. For example, soy-formula is positively associated with *Lachnospiraceae*, whereas breastfed infants have more *Bifidobacterium* and *Lactobacillus*.

Current guidelines do not contain diverse nutritional approaches or requirements for male and female human infants [175], although human male preterm neonates respond better than females to preterm-enriched formula to meet higher preterm requirements ([153] and cited literature).

The use of soy-based formulas without a real medical need should be carefully examined. Parents should be informed beforehand about the possible estrogenic effects of soy-based formula during infancy, even if its long-term health effects remain the subject of debate.

### 7.6. Developmental Trajectory

Roseboom et al. [176] provided evidence that different programming outcomes depend on the time of exposure during pregnancy trimesters, and in some cases they may depend on sex [177]. In other words, some sex–gender differences in adult life may origin during prenatal and neonatal life. At this regard, we recall that some placentation processes may be sex-specific [178,179,180]. Nowadays, it is clarified that the prenatal and postnatal life environments, including diet, changes determine both immediate and long-term effects that can be major factors in addressing the risk of adult disease, including susceptibility to develop obesity, diabetes, cardiovascular diseases, etc. [115,181]. Indeed, long-term consequences of modifications in the early life environment are linked with adult mortality and expected lifespan in men and women [177]. Interestingly and relevantly, these processes occur in a sexually-dimorphic way [115]. Early phytoestrogens exposure seems to be associated with the age of puberty; however, the direction of this link seems to depend on the single phytoestrogen [182].

Finally, assumption of licorice during pregnancy is related with a lower intelligence quotient, low memory, and elevated risk of attention deficit in the children [183]; however, it is not clear if these phenomena are influenced by the sex. The pubertal advancement observed in girls exposed to licorice during the gestational period could be ascribed to glabridin, a compound with estrogenic activity present in the licorice [183,184]. In particular, girls are taller and heavier, and have higher BMI for age, while cortisol levels do not diverge [183,184]. Neonatal exposure to relevant amount of isoflavones may affect female fertility in mice [185]. 

## 8. Conclusions

In view of the rapid increase in the global consumption of phenols, further insights into their risks and benefits on health seem essential; thus, additional studies with individuals of all ages and sex–genders are needed because the ratio risk/benefit is still unclear. Prescription drugs are marketed only after approval from regulatory authorities, which occurs after accurate preclinical research and after randomized clinical trials; this does not occur with the majority of marketed herbal products. In view of overcome health problems, these studies should be performed also for the botanicals. These studies must introduce sex-gender aspects in order to favor transition toward tailored nutrition and medicine. Indeed, to design a personalized medicine, it is necessary to investigate the role of the gut microbiota and to consider “Thrifty Phenotype Hypothesis” [21]; thus, the long-term effects of phenols, especially when exposure occurs during prenatal and neonatal life, must be known. This is especially true for phenols with hormonal activities. Finally, the social environmental should be considered in the design of studies.

## Figures and Tables

**Figure 1 nutrients-11-01637-f001:**
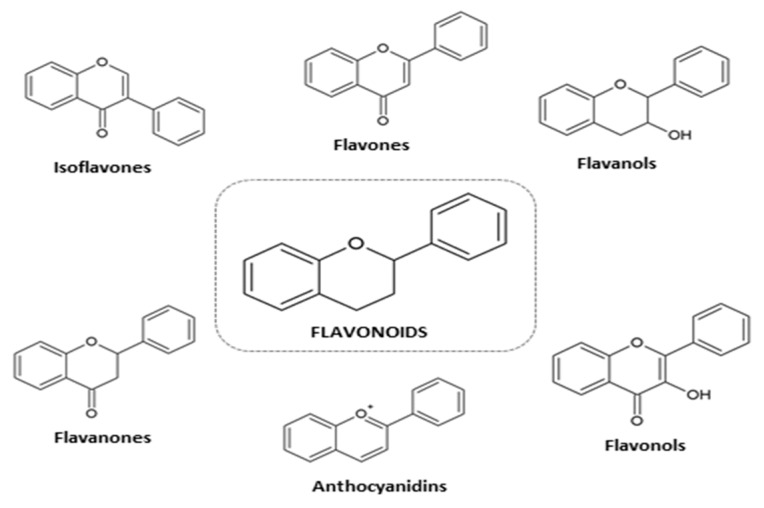
Chemical structure of the different classes of flavonoids.

**Figure 2 nutrients-11-01637-f002:**
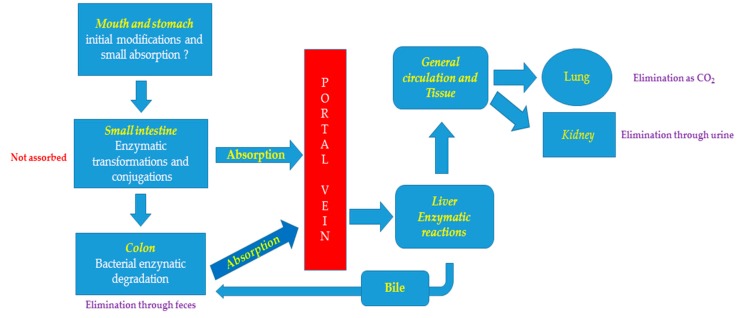
Schematic predicted routes for absorption and elimination of dietary phenols.

**Figure 3 nutrients-11-01637-f003:**
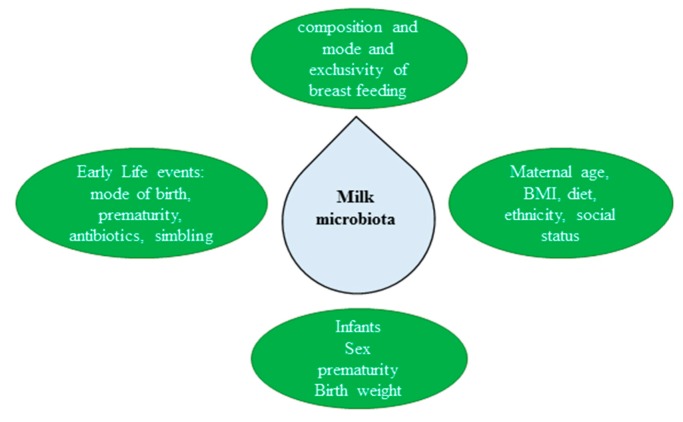
Factors that influence the milk human microbiota. BMI: body mass index.

**Table 1 nutrients-11-01637-t001:** Some sex–gender differences involved in pharmacokinetics.

Parameters	Male/Female
Fat	+F (human)
Muscular mass	+M (human)
Body weight	+M (human)
Height	+M (human)
Heart rate	−M (human)
Regional blood flow	+M (human)
Plasma volume	+F (human)
Total water	+M (human)
Gastric pH [acidity]	+M (human)
Gastric emptying	+M (human)
Gastro-intestinal mobility	+M (human)
Glomerular filtration rate	+M (human)
CYP1A2	+M (human)
CYP2A6	+F (human)
CYP2A7	+F (rat)
CYP2E1	+M (human)
CYP3A4	+F (human)
CYP3A	only in males (pig)
CYP3A5	+M (human)
CYP3A7	+F (human)
CYP3A9	+F (human)
CYP2B6	+M (human)
CYP2C9	= (human)
CYP2C19	= (human)
CYP2D6	+M (human)
CYP7A1	+F (human)
COMT	+M (human)
GST	+M (rat)
GSTA1/A2	+F (human)
UDP-glucuronosyl-transferases	+M [UGT1A6 (pig); UGT2b1 (liver); UGT2b5/37/38 (kidney); UGT1a6 (lung); UGT2b15; UGT2b17] +F [UGT 1a1 (human); UGT 1a5 (liver); UGT 1a2 (kidney); UGT 2b35 (brain)]
SULT1A1	+F than men with high androgen levels (human)
SULT1E1 liver	only in males (rat)
Oatp1	= (rat)
Oatp2	= (rat)
Oatp4	+F = (rat)
P-glycoprotein	+M (human)
Liver SLC3A1	+F (human)
Liver SLC13A1	+M (human)
Liver SLC10A1	+F (human)
Liver ACSL4	+F (human)

Data are obtained from ([40,41,42] and cited literature) F = female, M = male; Oatp = Organic-anion-transporting polypeptide, SLC = solute carrier family

**Table 2 nutrients-11-01637-t002:** Activity of some phenols and herbs on drug metabolic enzymes and transporters.

Species	Enzyme/Transporters	Phenols/Herbs	Type of Activity	References
Pig	CYP 1A1	*Genistein*	Inh	[6]
Pig	CYP 1A1	*Daidzein*	Inh	[6]
Pig	CYP 1A1	*Biochanin*	Ind	[6]
Pig	CYP 1A1	*Equol*	Inh	[6]
Pig	CYP 1A1	*Rutin*	Inh	[17]
Pig	CYP 1A1	*Myricetin*	Inh	[17]
Pig	CYP 1A1	*p-couamric acid*	Inh	[17]
Pig	CYP 1A1	*Gallic acid*	Inh	[17]
Pig	CYP 1A1	*Caffeic acid*	Inh	[17]
Human	CYP1A2	*Echinacea purpurea*	Inh	[186]
Human	CYP1A2	*Garlic oil*	=	[187,188]
Human	CYP1A2	*Allium sativum*	Inh	[189]
Human	CYP1A2	*Matricaria recutita*	Inh	[190]
Human	CYP1A2	*Gongronema latifolium*,	Inh	[189]
Human	CYP1A2	*Moringa oleifera*	Inh	[189]
Human	CYP1A2	*CG (-)-catechin-3-O-gallate, GCG (-)-gallocatechin-3-O-gallate, EGCG*	Inh	[191]
Human	CYP1A2	*Berberine*	Inh	[192]
Rat	CYP1A2	*Genistein*	Inh	[48] and cited literature
Human	CYP1A	*Mangifera indica*	Inh	[189]
Human cancer cell	CYP1A4	*Genistein*	Inh	[48] and cited literature
Human	CYP2B6	*Allium sativum*	Inh	[189]
Human	CYP2B6	*Mangifera indica*	Inh	[189]
Human	CYP2E1	*Garlic oil*	Inh	[187]
Human	CYP2E1	*Piper methysticum*	Inh	[188]
Human	CYP2E1	*St John’s wort*	Ind	[187]
Pig	CYP2E1	*Quercetin*	Inh (male)	[19]
Pig	CYP2E1	*Rutin*	Inh	[19]
Pig	CYP2E1	*Myricetin*	Inh	[19]
Pig	CYP2E1	*p-couamric acid*	Inh	[19]
Pig	CYP2E1	*Gallic acid*	Inh	[19]
Pig	CYP2E1	*Caffeic acid*	Inh	[19]
Rat	CYP2C	*Genistein*	Inh	[48] and cited literature
Human	CYP2C8	*Allium sativum*	Inh	[189]
Human	CYP2C8	*Mangifera indica*	Inh	[189]
Human	CYP2C9	*Mangifera indica*	Inh	[189]
Human	CYP2C9	*Allium sativum*	Inh (2c9*1) =	[193] [189]
Human	CYP2C9	*Matricaria recutita*	Inh	[190]
Human	CYP2C9	*(-)-epicatechin-3-O-gallate ECG, (-)- epigallocatechin, EGC CG (-)-catechin-3-O-gallate*,	Inh	[191]
Human	CYP2C9	*Berberine*	Inh	[192]
Human	CYP2C19	*Achillea millefolium*	Inh	[190]
Human	CYP2C19	*Ginkgo biloba*	Ind	[194]
Rat	CYP2D2	*Isorhamnetin*	Inh	[17,64]
Rat	CYP2D2	*Resveratrol*	Inh	[72,195]
Human	CYP2D6	*Hydrastis Canadensis*	Inh	[196]
Human	CYP2D6	*Allium sativum*	=	[189,193]
Human	CYP2D6	*Garlic oil*	=	[187,188]
Human	CYP2D6	*Cimicifuga racemosa*	Inh	[188]
Human	CYP2D6	*Mangifera indica*	Inh	[189]
Human	CYP2D6	*Alstonia boonei*	Inh	[189]
Rat	CYP2D6	*Alstonia scholaris*	Inh	[190]
Human	CYP2D6	*Matricaria recutita*	Inh	[190]
Human	CYP2D6	*Picralima nitida*	Inh	[189]
Human	CYP2D6	*Berberine*	Inh	[192]
Human	CYP3A4	*St John’s wort*	Ind Ind	[45] [187]
Human	intestinal CYP3A4	*Grapefruit juice*	Inh =	[197]
Human	liver CYP3A4 intestinal CYP3A4	*Echinacea purpurea*	Inh Ind	[186] [198]
Rat	CYP3A	*Genistein*	Inh	[48] and cited literature
Pig	CYP3A	*Myricetin*	Inh (male)	[17]
Human	CYP3A4	*Allium sativum*	Inh =	[193] [199]
Human	CYP3A4	*Matricaria recutita*	Inh	[190]
Human	CYP3A4	*Picralima nitida*	Inh	[189]
Human	CYP3A4	*Achillea millefolium*	Inh	[190]
Human	CYP3A4	*CG (-)-catechin-3-O-gallate, GCG (-)-gallocatechin-3-O-gallate, EGCG*	Inh	[191]
Human	CYP3A4	*Berberine*	Inh	[192]
Human	CYP3A5	*Allium Sativum*	Inh =	[193] [199]
Sheep	BCRP	*Genistein*	Ind	[45]
Human	Oatp1A2	*Green tea extract*	Inh	[200]
Human	P-glycoprotein/MDR1	*Garlic extract*	Ind	[199]
Human	P-glycoprotein/MDR1	*St John’s wort*	Inh	[45]
Rat	P-glycoprotein/MDR1	*St John’s wort*	Inh	[45]
Human cancer cell line	UDP	*Genistein*	Ind	[48] and cited literature
Human	SULT	*Genistein*	Inh	[48] and cited literature
Human	COMT	*Epigallocatechin-3-gallate*	Inh	[68]

Inh = inhibition; Ind = induction.

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
