# Peer review of "The Sex–Gender Effects in the Road to Tailored Botanicals"

_nutrients, 2019, doi:10.3390/nu11071637_

Round 1
Reviewer 1 Report
In this paper, Campesi and colleagues have performed an extensive review of the literature on the influence of sex/gender on phenolic compounds' activities. This is a very interesting topic and their major ideas specified in the conclusions are relevant. However, the authors have recently published similar articles in this field with similar (sometimes overlapping) content, and the structure of the current manuscript is not easy to follow due to the extensive number of articles included and sometimes the lack of connection/flow between the different studies.
Although the review is detailed, I consider that it should be improved in several aspects.
- The statement found in the abstract that the beneficial effects of phenols are rarely evidenced in clinical practice "due to their chronic toxicity" is not accurate.
- The abstract should be re-written as it does not mention that this is a review article, nor which is the scope of this review.
- Most important point: The same authors have a recent review published 6 months ago in the European Journal of Nutrition which partially overlaps the contents of the present article. Some figure and even some whole sentences are very similar. They should perform major modifications of the present article and stating only what is the novelty of this review article.
- The authors have used long paragraphs that at some point are very hard to read. Example, on page 11 the paragraph starting in line 385 and finishing in 421 (36 lines mixing information from clinical studies, preclinical studies, and different bioactive compounds).
- In line 355, reference 21 after "nowadays" referring to an article published in 1991 does not make much sense.
Author Response
Referee 1
In this paper, Campesi and colleagues have performed an extensive review of the literature on the influence of sex/gender on phenolic compounds' activities. This is a very interesting topic and their major ideas specified in the conclusions are relevant. However, the authors have recently published similar articles in this field with similar (sometimes overlapping) content, and the structure of the current manuscript is not easy to follow due to the extensive number of articles included and sometimes the lack of connection/flow between the different studies.
Now numerous parts of manuscript have been rewritten in order to ameliorate the flow among different studies
Although the review is detailed, I consider that it should be improved in several aspects.
- The statement found in the abstract that the beneficial effects of phenols are rarely evidenced in clinical practice "due to their chronic toxicity" is not accurate. Now it has been modified ,
The abstract should be re-written as it does not mention that this is a review article, nor which is the scope of this review. We thank the referee for her/his suggestion, the abstract has been modified accordingly.
-Most important point: The same authors have a recent review published 6 months ago in the European Journal of Nutrition which partially overlaps the contents of the present article. Some figure and even some whole sentences are very similar. They should perform major modifications of the present article and stating only what is the novelty of this review article.
The previous review was focused on cardiovascular and antioxidant effects of phenols and these topics include 3 sections (precisely: Effect on vascular function, Effect on CVD prevention, Oxidative stress among males and females).. Obviously it has been also discussed the sex differences in regulation of redox state in males and females. Finally, the section titled “Nuclear Receptors” is focused only on estrogen receptors. All these arguments are not present in the submitted review. Whereas, the submitted review present 7 sections (precisely “sex differences in plants”, “interactions”, “pregnancy”, “lactation”, “galactagogues based on herbs”, “breast milk”, “soy-milk formula”) that are not present in the paper published J Nutr. 2018
Further, in J Nutr. 2018 Dec; 57 (8): 2677-2691, the introduction was based on CVD and redox state and on influences of sex and gender on them, here the introduction is focused on other aims.
The section “Phenolic compounds classification” (in the previous review) and “plant materials” (in this review) reports the classification of phenols. In our opinion, according with others (Nutrients 2016, 8(2), 78; Nutrients. 2019 Jun 16;11(6), Antioxidants (Basel). 2019 Jun 6;8(6), the description of phenol structure (that in this new review is very short) is indispensable prerequisites for discussing phenol activity and help the reader to fully understand the other review sections.
Regarding the section “Sex dependent-bioavailability ….” of the review published on Eur J Nutr. We underlie that it is mainly concentrated on metabolism of phenols. Here, instead we dedicate space to all the phases of the pharmacokinetic therefore we believe that changing the title helps the reader. The discussion on distribution, absorption, elimination evidences how the pregnancy may modifying them. The changes induced by pregnancy on the absorption, metabolism, distribution, elimination are necessary to explain to the reader that pregnant and not pregnant women are not similar when pharmacokinetic and pharmacodynamic of phenols are studied. Finally, this section are indispensable to understand the section “Interactions”.
Obviously, it is not possible to discuss the pharmacokinetic aspect without considering the gut microbiota, here we also emphasized milk microbiota, which was absent in the previous paper, and the change induced by lactation and pregnancy on gut microbiota.
In comparison with previous work an update of literature has been performed.
- The authors have used long paragraphs that at some point are very hard to read. Example, on page 11 the paragraph starting in line 385 and finishing in 421 (36 lines mixing information from clinical studies, preclinical studies, and different bioactive compounds). The section has been modified, trying to report information in a more orderly manner.
- In line 355, reference 21 after "nowadays" referring to an article published in 1991 does not make much sense. We agree with the referee observation, the term "nowadays" has been deleted, as inappropriate

Reviewer 2 Report
The review by Campesi et al brings an important contribution to the scientific community giving the increasing need to consider gender/sex as key variable in the pharmacotherapy.
Author Response
We thank the referee for his/her positive analysis of our manuscript
Round 2
Reviewer 1 Report
The authors have correctly addressed all the questions raised in the previous report in which the reviewer recommended major changes. The manuscript is now clearer and only very minor modifications are suggested in order to improve the quality of the manuscript:
1. Abstract: the words “due to their potential and not fully understood toxicity” may confuse the reader as the paper does not focus directly on toxicity but rather on the influence that sex/gender may have on the effects of phenolic compounds. The reviewer completely agrees with the fact that phenolic compounds have a multitude of biological actions and that their beneficial effects are not commonly evidenced in clinical practice but this is due to the lack of clinical trials evaluating their effects (and not strictly due to their potential and not fully understood toxicity).
2. The authors use several words like “phenolic compounds”, “xenobiotics”, “phenolics”, “botanicals” and being more homogeneous could help the reader.
3. On page 2, please review the classification of the different families of flavonoids as according to the current sentence it seems that catechins are flavanols when they are not (they are flavanols). Additionally, add a reference of an article or review on this topic.
4. On page 4 and 9, please substitute the word “pharmacokinetic” by “pharmacokinetics”
5. The expression [X reference “and cited literature”] is often employed throughout the manuscript and it seems ambiguous. I would recommend keeping just the relevant references with their corresponding numbers.
6. Last recommendation to improve the manuscript: differentiate (by separating in different paragraphs) between those sex-gender differences observed in animal models from the sex-gender differences observed in humans.
Author Response
1. Abstract: the words “due to their potential and not fully understood toxicity” may confuse the reader as the paper does not focus directly on toxicity but rather on the influence that sex/gender may have on the effects of phenolic compounds. The reviewer completely agrees with the fact that phenolic compounds have a multitude of biological actions and that their beneficial effects are not commonly evidenced in clinical practice but this is due to the lack of clinical trials evaluating their effects (and not strictly due to their potential and not fully understood toxicity). Now, the sentence was deleted to avoid confusion
2. The authors use several words like “phenolic compounds”, “xenobiotics”, “phenolics”, “botanicals” and being more homogeneous could help the reader. As suggested, phenolics compounds and phenolics are substituted with phenols. Some “xenobiotics” and botanicals are left when they include other molecules beyond phenols
3. On page 2, please review the classification of the different families of flavonoids as according to the current sentence it seems that catechins are flavanols when they are not (they are flavanols). Additionally, add a reference of an article or review on this topic. The point is now corrected
4. On page 4 and 9, please substitute the word “pharmacokinetic” by “pharmacokinetics” The point is now corrected
5. The expression [X reference “and cited literature”] is often employed throughout the manuscript and it seems ambiguous. I would recommend keeping just the relevant references with their corresponding numbers. The limit in the number of references (100). the 100 references limit prevented us from quoting the individual authors; so we had to use the reviews, but with the formula adopted and cited literature, we believed and believe to give the right value to the authors not mentioned individually.
6. Last recommendation to improve the manuscript: differentiate (by separating in different paragraphs) between those sex-gender differences observed in animal models from the sex-gender differences observed in humans. In table 2, the animal species has always been reported, now the animal species is always reported also in table 1. In our opinion, this highlights the importance of the species without overload the text